# Microsurgical Treatment of Arteriovenous Malformations: A Single-Center Study Experience

**DOI:** 10.3390/brainsci13081183

**Published:** 2023-08-10

**Authors:** Ali Abdi Maalim, Mingxin Zhu, Kai Shu, Yasong Wu, Suojun Zhang, Fei Ye, Ying Zeng, Yimin Huang, Ting Lei

**Affiliations:** Department of Neurosurgery, Tongji Hospital, Tongji Medical College, Huazhong University of Science and Technology, Wuhan 430030, China; alimaalim1@hotmail.com (A.A.M.); mxzhu@tjh.tjmu.edu.cn (M.Z.); kshu@tjh.tjmu.edu.cn (K.S.); yswu@tjh.tjmu.edu.cn (Y.W.); zhangsuojun@tjh.tjmu.edu.cn (S.Z.); feiye@tjh.tjmu.edu.cn (F.Y.); tuzi76@126.com (Y.Z.); yimin.huang@tjh.tjmu.edu.cn (Y.H.)

**Keywords:** microsurgery, arteriovenous malformation, AVM, unruptured AVM, ARUBA-eligible

## Abstract

Objective: The purpose of the study was to assess the functional outcomes after microsurgical resection of arteriovenous malformations (AVMs) and to compare the results between patients eligible for A Randomized Trial of Unruptured Brain Arteriovenous Malformations in this surgical series to the results reported and the ARUBA study. Methods: We reviewed the records of 169 patients who underwent microsurgical treatment of arteriovenous malformation (AVMs) in our institution between January 2016 and December 2021. These patients’ functional status was assessed using modified Rankin Scale (mRS) scores at the last follow-up and before treatment. The mRS scores at the latest follow-up were classified into good outcomes (mRS < 3) and poor outcomes (mRS ≥ 3). Clinical presentation, patients’ demographics, AVM characteristics, follow-up time, and obliteration rate were analyzed. Subgroup analyses were performed on the whole cohort, comparing Spetzler–Martin Grade I and Grade II, and ARUBA-eligible AVMs. Results: The initial hemorrhagic presentation occurred in 71 (42%) out of 169 patients. The majority of the patients presented with headaches (73%). The AVMs were completely obliterated in 166 (98.2%) patients. The series included 65 Spetzler–Martin Grade I (38.5%), 46 Grade II (27.2%), 32 Grade III (18.9%), 22 Grade IV (13%), and 4 Grade V (2.4%) AVMs. There were 98 unruptured and 79 ARUBA-eligible cases. Also, optimal functional outcome was achieved in 145 (85.8%) patients. The overall mortality rate was 5.3% (9/169). The multivariate analysis illustrated that a poor outcome was significantly associated with presurgical mRS ≥3 (*p* < 0.013; OR, 0.206; 95% CI 0.059–0.713), increasing age (*p* < 0.045; odds ratio [OR], 1.022; 95% CI 1.000–0.045), and female gender (*p* < 0.009; OR, 2.991; 95% CI 1.309–6.832). Conclusions: Our study suggests that better outcomes can be obtained using microsurgical resection in the majority of patients with AVMs. Independent predictors of poor outcomes after surgical resection of AVMs include increasing age at the time of surgery, poor presurgical functional status, and female gender. Supposing that patients are more suitable for microsurgery after presurgical examination, outcomes are normally better in that case than those achieved by multimodal interventions (such as conservative treatment or ARUBA treatment arm). Therefore, we recommend early surgical removal on all surgically accessible AVMs to prevent successive hemorrhages and the consequences of poor neurological outcomes.

## 1. Introduction

The brain’s arteriovenous malformations (AVMs) are mainly composed of tangles of poorly differentiated cerebral arteries and veins that merge at a vascular nidus without normal adjoining parenchyma. Parenchymal AVMs have often been regarded as congenital lesions. This notion has been refuted by multiple accounts of de novo AVM development and the proof that parenchymal brain AVMs (except for the vein of Galen AVMs) are never seen on prenatal ultrasonography [1]. Brain AVM incidence and prevalence are still poorly understood between 5 and 613 incidences per 100,000 [2]. The estimated frequency of AVMs in autopsy investigations varies significantly. The total incidence of AVMs varies among population-based studies and is between 1.10 and 1.42 cases per 100,000 [3,4].

Despite considerable advances in the last three decades with the debut of radiosurgery and microsurgery, treating cerebral arteriovenous malformations (AVMs) remains controversial. Even though AVMs are rare, they are associated with a life-long risk of stroke due to rupture, which has been reported to occur somewhere between 2% and 4% yearly [5].

While it is generally accepted that ruptured AVMs must be removed entirely or cauterized, the management of unruptured AVMs has been the subject of extensive review and discussion in the light of the recent ARUBA (A Randomized Trial of Unruptured Brain Arteriovenous Malformations) study, which compared medical management to interventional therapy for unruptured AVMs [6]. It shows that medical care is superior to interventional treatment for preventing mortality or stroke in individuals with unruptured brain AVMs. This requires thoroughly scrutinizing the interventions for unruptured AVMs [7]. Only 5 of the 322 patients received microsurgical care alone; 13 of the 322 patients received microsurgical care as a component of multimodal therapy. Even though prospective randomized clinical trials are, in most cases, regarded as being superior to single-center series, some authors criticized the small number of patients who underwent microsurgical resection alone and concluded that two treatment modalities (embolization and radiosurgery) known for a high rate of incomplete AVM occlusion led to the result that interventional treatment might be easily interpreted as making microsurgery ineffective [8,9,10,11,12,13,14,15]. Later, emphasis was mainly based on lower Spetzler–Martin grades, where it was found that outcomes were better. Attention was also focused on subgroups that would have been eligible for the ARUBA trial if they had met its eligibility requirements. Several series of ARUBA-eligible cases have been published, demonstrating that microsurgery is still an effective treatment option for certain subgroups [6,8,11,16,17].

This patient series includes all AVMs surgically removed at our institutions between 2016 and 2021. This study aims to contribute data on outcomes after microsurgery, including unruptured AVMs, ARUBA-eligible, and Spetzler–Martin Grades I and II.

## 2. Methods

### 2.1. Inclusion and Exclusion Criteria

Figure 1 depicts the general enrollment process of patients with AVM into our study. Accordingly, there were a total of 477 patients who were diagnosed with AVM; after fulfilling ultimate or extended criteria, we excluded 308 patients who had the following features: endovascular embolization (*n* = 134), combined surgery (*n* = 18), AVM with aneurysms (*n* = 26), no surgery (*n* = 109), and detailed data were unavailable (*n* = 21). After excluding the aforementioned cases, 169 AVM were maintained for subsequent descriptive, comparative, and logistic regression analysis.

### 2.2. Patients and Study Design

This study was approved by the Institutional Review Board of Tongji Hospital Affiliated to Huazhong University of Science and Technology. We conducted a retrospective analysis of departmental and hospital databases with approval from the Board Review to identify all patients diagnosed with AVMs between January 2016 and December 2021. Complete radiological and clinical records for this period were accessible from all databases. All previous follow-up notes and charts were reevaluated and reviewed. Patient age, BMI, sex, clinical presentations, Spetzler–Martin grade and location of the AVM, size, hemorrhage, mortality rate, recurrent rates, estimated surgical blood loss, and neurological complications were all factors we looked for in the patient records.

### 2.3. Statistical Analysis 

Statistical analyses were performed using SPSS version 25. AVM characteristics, and patient demographics were encapsulated using descriptive statistics for categorical variables (count and percentage) and continuous variables (mean standard deviation). In analyses of neurological outcomes, the patients were divided into two groups: those with poor outcomes and those with good outcomes. Univariate analyses were used to compare variables between the two groups. These variables included patient demographics (age, sex, body index mass, blood group, initial presentation, preoperative mRS, hospital stay, and follow-up period) and AVM characteristics (AVM location, size, duration of surgery, blood loss, and Spetzler–Martin grade). A logistic regression model was used to evaluate predictors of a poor neurological outcome at the last follow-up. To that end, univariate analysis was used to assess each variable first. All variables that were significant in the univariate analysis were retained for the multivariate analysis to separate the factors independently associated with poor neurological outcomes. A two-tailed *p* < 0.05 was regarded as statistically significant. Odd ratios (OR) and 95% confidence intervals (95% CI) were calculated to estimate the relative risk of poor neurological outcomes. Multivariate logistic regression analyses were used to evaluate predictors of poor neurological outcomes at the last follow-up. In each analysis, a 2-tailed *p* < 0.05 was regarded as statistically significant.

## 3. Results 

### 3.1. Baseline Features of 169 Patients with Surgically Treated AVMs

Table 1 shows the baseline characteristics of the 169 patients at the initial presentation. In this study, the male-to-female ratio was 63:36, and the average age at surgery was 35.04 ± 19 years (1–74 years). 

The initial hemorrhagic presentation occurred in 42% of the patients. Most of the patients presented with headache (73%), and the remaining patients had neurological dysfunction (20%), seizures (14%), loss of consciousness (17%), visual problems (8%), dizziness and vomiting (30%), memory loss (2%), and an asymptomatic/incidental finding (5%). Of all patients, 43 (25.4%) presented with a presurgical mRS ≥ 3, and 126 presented with a presurgical mRS < 3. AVM locations were as follows: frontal lobe, 24.3%; temporal lobe, 26.6%; parietal lobe, 7.1%; occipital, 16%; cerebellar, 17.8%; frontoparietal lobe, 3%; frontotemporal lobe, 3.6%; and parieto-occipital lobe, 1.8%. Most AVM sizes were <3 in 107 (63.3%) patients and ≥3 cm in 62 patients. Body index mass (BMI) was categorized into four groups: underweight, 19.5% (<18.5); normal, 57.4% (18.5 ≤ 25); overweight, 20.1% (25 ≤ 30); and obese, 3% (>30). In the blood group: 59 of the patients were blood group A (34.9%), followed by blood group O with 51 patients (30.2%), AB blood group with 16 patients (9.5%), and blood group B with 43 patients (25.4%). Classified by Spetzler–Martin grade [18], 38.5% of the AVMs were Grade I, 27.2% were Grade II, 18.9% were Grade III, 13% were Grade IV, and 2.4% were Grade V.

### 3.2. Surgical Technique 

In each case, a craniotomy was carried out to ensure that the lesion’s margins were well within the limits of the surgical exposure; before the dura was opened, we made sure there were matching blood products, a draping microscope, and in some cases, temporary aneurysm clips. The nidus might be located when feasible by following a superficial draining vein discovered early on. Then, coagulation occurred at prominent artery feeders that were easy to identify. Before disconnecting the venous drainage, an anatomical circumferential cortical and subcortical dissection was carried out with serial coagulation of the supplying artery pedicles. For patients with hemorrhages, hematoma evacuation with AVM resection was performed at the same time to minimize the expense rather than performing two separate surgeries at different times. The use of microslips on occasion for fragile and deep white matter feeders or draining veins was required by AVMs that extended deep into the white matter. To assist in separating arterial feeders from arterialized veins, locating en route vessels, and confirming resection, indocyanine green angiography was used.

### 3.3. Presurgical Workup 

All patients suspected of AVM were routinely examined by cerebral angiography (DSA) to make a definite diagnosis. Then, the Cerebrovascular Committee of our institution reviewed the treatment plan concerning the current worldwide standards. The final treatment plan was based on the patient’s demographic characteristics, symptoms, complications, and radiological results, including the size and location of AVM, the number of blood supply arteries, and clinical manifestations. After discussing each case at the neuroradiology meeting, neurosurgeons and neuroradiologists agreed on the best course of action. All patients underwent microsurgery. 

### 3.4. Postoperative Management 

All patients received standard follow-up and evaluation of nervous system function after the operation. The incidence of bleeding after surgery, the complete disappearance of the disease as shown by imaging examination, (Figure 2 and Figure 3) and the existence of new and lasting neurological damage are the criteria for the results. 

For patients with good hemostatic effects, the catheter is usually pulled out early and they are transferred from the intensive care unit. For patients with poor hemostatic effects, monitoring them in the intensive care unit for one night is generally necessary, controlling the maximum systolic pressure ≤100 mm Hg and then letting them wake up the next day. These patients were transferred to the ward after 24 h of routine observation in the intensive care unit. All lesions were reviewed by angiography one week after surgery and in the following years (usually shortly thereafter).

### 3.5. Outcome Evaluation 

The interval between microsurgical treatment and the final follow-up was designated the follow-up period. The average time of follow-up was 41.01 ± 21.46 months. The following factors were used to assess treatment outcome: obliteration rate, treatment-related complications, and mRS at the final follow-up. Obliteration rates were determined using postoperative DSA or CTA, often performed 3–7 days after surgery. Following surgery, treatment-related complications were noted. A mRS score of <3 indicated a good neurological outcome, whereas a score of ≥3 indicated a poor outcome.

### 3.6. Surgical Results

Postoperative DSA demonstrated that the AVMs had been completely obliterated in 98.2% (166/169) of patients. DSA identified residual AVM in three patients; two had surgery, while one opted for conservative treatment. The overall estimated mean volume of blood loss that occurred during AVM surgery was 546.92 ± 703.03. At our last follow-up, the residual AVMs in the two patients had been obliterated, while the other patient was doing well. Overall, six (3.6%) patients died after elective surgery (two patients) or emergency surgery (four patients). During our follow-up period, three more patients passed away. Five patients developed postoperative bleeding, three requiring hematoma evacuation and decompressive craniectomy. The five patients who experienced postoperative hemorrhages had a good neurological outcome at the final follow-up. Ten patients had a post-operative intracranial infection but recovered quickly following antibiotic treatment.

### 3.7. Long-Term Neurological Outcome 

In this surgical series, an overall satisfactory result (defined as a final mRS score < 3) was obtained from 85.8% of the patients (145/169) after a mean follow-up period of 41.01 ± 21.46 months. The overall mortality rate was 5.3% (9/169). In terms of the functional outcomes according to the surgical modalities (emergency surgery or elective surgery), a good outcome was achieved in 89.1% of patients who underwent elective surgery as their first surgical modality, and in 54.8% of patients who underwent emergency surgery to evacuate a life-threatening hematoma plus AVM resection. 

### 3.8. Predictors of Poor Outcome after Surgical AVM Resection

Table 2 summarizes the demographic and clinical parameters associated with post-microscopic surgery outcomes in individuals with arteriovenous malformation. Accordingly, all patients were divided into two groups (good outcome vs. poor outcome). A total of 145 (85.8%) had good outcomes, whereas 24 (14.2%) had poor outcomes. 

As applicable, we compared the demographic and clinical characteristics of the good and poor outcome groups using a Chi-squared independent test or Fisher’s exact test and student *t*-test. As a result, there was a significant difference in the AVM-associated parameters such as age (*p* = 0.017), sex (*p* = 0.006), presurgical mRS (*p* = 0.002), and venous drainage (*p* = 0.009). No significant difference was observed concerning the patient body index mass (*p* = 0.384), blood group (*p* = 0.403), presentation (*p* = 0.322), AVM size (*p* = 0.585), grading (*p* = 0.403), location (*p* = 0.944), hemorrhage (*p* = 0.629), duration of surgery (*p* = 0.756), blood loss (*p* = 0.237), arterial blood supply (*p* = 0.099), recurrent rate (*p* = 0.706), or follow-up (*p* = 0.677), and the difference in duration of hospital stay was only marginally significant (*p* = 0.049). The predictive impact of risk factors for patients with AVM following microscopic surgery was investigated using logistic regression analysis as depicted in Table 3. 

Initially, the role of each variable was examined separately for the risk factor outcome in a univariate analysis. As a result, a significant difference in risk factors was obtained in association with age (*p* = 0.014), sex (*p* = 0.037), presurgical mRS (*p* = 0.003), venous drainage (*p* = 0.020), and arterial supply (*p* = 0.012). Finally, a multivariate logistic regression analysis (Table 3) was conducted, in which all factors were considered to investigate further the factors that independently predict risk factors. Thus, increasing age (odds ratio [OR], 1.022; 95% confidence interval [95% CI], 1.000–0.045; *p* = 0.045), female gender (odds ratio [OR], 2.991; 95% confidence interval [95% CI], 1.309–6.832; *p* = 0.009), presurgical mRS score ≥ 3 (OR, 0.206; 95% CI, 0.059–0.713 *p* = 0.013). The presence of deep venous drainage (OR, 0.318; 95% CI, 0.093–1.090 *p* = 0.068) and double arterial supply (OR, 2.226; 95% CI, 0.934–5.306 *p* = 0.071) did not show any significant contribution to risk factors after multivariate analysis.

### 3.9. Hemorrhagic Versus Non-Hemorrhagic Presentation

Almost sixty percent of patients (*n* = 98) presented with a hemorrhage. Compared to the unruptured cases, this group contained considerably more Grades I and II AVMs, significantly fewer Spetzler–Martin Grades IV and V AVMs and were found to have more deep venous drainage. The incidence of the new deficit was much higher in the unruptured group (9.2% vs. 4.3%). Similarly, persistent new deficits occurred at a higher rate (3.1%) than the hemorrhagic group (2.8%).

### 3.10. Subgroup Eligible for ARUBA Study

A total of 79 cases, as shown in the Table 4, including 24 Spetzler–Martin Grade I, 32 Spetzler–Martin Grade II, 18 Spetzler–Martin Grade III, and 5 Spetzler–Martin Grade IV AVMs, were found to meet the inclusion criteria for the ARUBA study, which excluded pediatric patients and patients with a pre-existing deficit. Many patients (87.3%) were found to have superficial venous drainage, and 36.7% had diameters ≥ 3 cm. All 79 patients had pre-operative mRS scores <3. In 3.8% of cases, a new deficit and a persistent new deficit was observed. There was no treatment-related death in the ARUBA subgroup. The ARUBA-eligible Spetzler–Martin Grades I and II subgroup included 56 patients. Table 5 demonstrates the deficit rates of the study population in comparison to the deficit rates of different subgroups.

### 3.11. Non-Hemorrhagic Subgroup

Out of a total of 111 Spetzler–Martin Grade I or II AVMs, there were a total of 68 that were unruptured in the combined Spetzler–Martin Grades I and II AVMS. These contained a total of 39 Spetzler–Martin Grade II AVMs in addition to 29 Spetzler–Martin Grade I AVMs. In some patients, a new post-operative deficit was recorded; however, persistent new deficits were reported in fewer cases, and there were zero instances of death. Table 6 displays the Spetzler–Martin Grades I and II group’s deficit rates. New deficits and persistent new deficits were more common in unruptured cases compared to hemorrhagic cases of the same Spetzler–Martin grades.

## 4. Discussion 

It has been hypothesized that arteriovenous malformations arise when primitive vascular channels fail to differentiate into arteries, intervening capillaries, or veins at the appropriate time in development [19]. In the cause of Julius Caesar’s epilepsy, Nicola Montemurro suggests that if AVM caused it, it is possible that Caesar’s vertigo, sensory loss, limb paresis, and gait disruption were caused by a common AVM steal phenomenon rather than a rarer embolic stroke or delayed ischemic stroke caused by spontaneous thrombosis [20]. The steal phenomenon, in which blood is directed preferentially to the AVM at the expense of normal brain parenchyma, can cause focal neurological symptoms like those Caesar experienced, as well as seizures, personality changes, irrational behavior, and, in extreme cases, focal atrophy [20]. Vera CL, in his study of “dual pathology” and the significance of surgical outcome in “Dostoevsky’s Epilepsy”, explains the auras of a unique experience of “love and union” with the physical realm, also described as “ecstasy”, often with mystical significance for the patient [21]. The purpose of the study was to present the results of microscopic surgery as the first-line treatment approach for arteriovenous malformations in a single center. The surgeons began to concentrate on surgical interventions for Spetzler–Martin Grades I to III early in the treatment process. Only lately has there been debate over whether or not arteriovenous malformations (AVMs) of Grade III should be categorized differently from lesions of Grades I and II [22,23]. An arteriovenous malformation (AVM) treatment options include endovascular embolization, radiosurgery, microsurgical resection, and any combination of two or three procedures. According to our experiences, we recommend having surgical resection performed on almost all AVMs that may be accessible by surgical operations to achieve complete obliteration and avoid subsequent hemorrhages.

### 4.1. The ARUBA Study 

In the ARUBA study, conservatively managed cases had a considerably lower stroke and mortality rate (10.1% vs. 30.7%). This was because only a minority of the patients underwent microsurgery alone, and a small group underwent a combination of microsurgery and embolization. The findings of a Scottish population-based study supported the lower incidence for conservatively managed AVMs [24]. However, when the primary outcome was examined more closely, it was found that the outcomes for both treated and untreated patients were similar after four years of following up. According to Russin and Cohen-Gadol, “Many of the outcome data in the treatment arm of the ARUBA trial are contaminated by previous therapeutic that has since been long abandoned because of low obliteration rates and complication rates”. This is because the treatment group in the ARUBA study is more reflective of the different non-microsurgical treatment methods and their side effects [12]. The ARUBA study only addresses unruptured AVMs. Hence, it is inappropriate to make conclusions about microsurgical therapy from it. The common belief that the ARUBA study has resulted in a major setback for microsurgery for AVMs, in general, is erroneous.

### 4.2. Microsurgical Outcome for AVMs 

Our study shows that 98.2% of AVMs were successfully removed, and 85.8% of patients had a favorable outcome. Mortality rates were 5.3% overall, and surgical mortality was 3.6%. The findings of our study were consistent with the previous findings. The study that was conducted by Rodriguez-Hernández on 60 surgically treated cerebellar AVMs found a 100% obliteration rate, a good outcome in 74% of cases, a 5% surgical mortality rate, and a 10% overall mortality rate [25]. Complete AVM resection was obtained in 92% of patients with posterior fossa AVMs, excellent and good results were achieved in 71% of patients, the surgical mortality rate was 15%, and the morbidity rate was 21% in the study by Drake. Other studies that reported on surgically treated posterior fossa AVMs have shown an 80–91% success rate, a morbidity rate of 9.0–17%, and a surgical mortality rate of 4.1–8.3% [26,27,28,29]. According to these findings, microsurgical resection should be the treatment of choice for most arteriovenous malformations (AVMs).

### 4.3. Comparing Spetzler–Martin Grade Outcomes

Fabio A. Frisoli et al., in his study, reevaluated SM Grade III AVMs to assess if the modified SM grade with its pluses, minuses, and asterisks, or the Supp-SM grade with its mixture of SM and LY rating systems, should be used to guide surgical selection for these patients. The findings strongly support using the Supp-SM grading system for such decisions [30].

Lawton et al. noted the variability of Spetzler–Martin Grade III AVMs. They proposed that they should be further identified as low-risk and high-risk lesions, suggesting more customized recommendations for Spetzler–Martin Grade III lesions [31,32,33]. Spetzler et al. and Ponce et al. also discovered that the risk profiles for Spetzler–Martin Grades I and II AVMs are identical but considerably different from those of Spetzler–Martin Grade III AVMs, which our study confirmed [33]. Several possible risk factors were found that may account for the observed variations in outcomes: twenty-nine of the ninety Spetzler–Martin Grade III AVMs were small, had deep venous drainage, with an eloquent location. The remaining sixty-one Spetzler–Martin Grade III lesions were greater than 3 cm, also linked with increased risk. Lawton et al. revealed that, among Spetzler–Martin Grade III lesions, morbidity increases when the size is more than 3 cm and dramatically increases when the location is eloquent, which is also observed in our study in terms of risk factors [33]. 

In this series, when looking at the complication rate, new deficits are 6.2% and 10.9% of Spetzler–Martin Grades I and II, respectively, and this percentage rises to 15.6% for Spetzler–Martin Grade III lesions (Table 5). The difference between the persistent new deficit for Spetzler–Martin Grades I and II versus Spetzler–Martin Grade III is higher (as shown in the table); similar results were reported by Johannes Schramm [15]. Steiger et al. found a significantly lower rate of permanent deficit in their series of 69 Spetzler–Martin Grades I and II AVMs compared to their reported series of 22 Spetzler–Martin Grade III lesions [17]. 

Considering the persistent new deficits for Spetzler–Martin Grades I and II vs. Spetzler–Martin Grade III lesions in this study, it is worth debating whether unruptured Spetzler–Martin Grade III lesions should be frequently considered for microsurgery. Morgan had previously arrived at a similar conclusion, lately supported by El Hammady and Heros [22,23].

### 4.4. The ARUBA-Eligible Group’s Outcome

Numerous studies that operated on AVMs that met the ARUBA criteria have clearly shown that, at least for Spetzler–Martin Grades I and II AVMs, patients who receive microsurgical resection of the AVM had better results than those who were treated with observation [8,16,34]. Our study supports these findings. Mortality rates were zero in both ARUBA-eligible Spetzler–Martin Grades I and II groups, but the persistent new deficit rate was 3.6%. These percentages are lower than those seen in the ARUBA study’s interventional and observational arms, where the risk of stroke or death was 10.1% and 30.7% for conservative and interventional treatment, respectively. Despite this ongoing debate, several authors recommend microsurgery as the best option for treating Spetzler–Martin Grades I and II AVMs [10,11,12,14,17,22,34].

### 4.5. Ruptured Versus Unruptured AVM

In our study, 42% of the patients had a ruptured AVM. In 31 patients, emergency surgery was carried out to remove a life-threatening hematoma plus AVM resection at the same time; the primary objective of this was to reduce cost and the level of anxiety due to second surgery for the patients and the family, as well as it potentially becoming another factor that affects patients’ recovery. Some patients might even experience having more than two surgeries depending on the age of the patient and the nature of the AVMs; some patients might end up having more surgeries, including decompressive craniectomy due to elevated intracranial pressure, which will later require cranioplasty, and permanent ventriculoperitoneal cerebrospinal fluid shunting will be needed in case of hydrocephalus. Combing all these factors will burden the patients’ families, so we recommend removing any hematoma plus AVM resection at the same time if there are no contradictions. The study by Xiangzeng Tong reported that neurological outcomes did not vary between individuals who had AVMs removed at the time of hematoma evacuation and those who had AVMs removed at a deferred time [35]. For patients without a life-threatening hematoma, we suggest removing the AVM after hemorrhagic presentation subsides, depending on the nature of the hemorrhagic manifestations. Based on our findings, we can conclude that 86% of our cases had a good outcome.

In this regard, we also recommend microsurgical resection for practically all AVMs except those with contraindications and those in patients older than 65. Most of the patients in our study with high S–M grade AVMs presented with ruptured AVMs. A good outcome is still possible for 52.4% of patients with ruptured high-grade AVMs. We suggest early surgical resection to achieve total obliteration and avoid recurrent hemorrhage from ruptured high S–M grade AVMs. Most unruptured AVMs in our surgical series were of S–M Grades I to III; this represents 58% of all cases. A good outcome was achieved in 91.2% after AVM resection.

The rates of complications for unruptured AVMs were higher than those for the overall population for Spetzler–Martin Grades I and II. The neurological decline was usually severe initially, but it eventually leveled out. Similar to what Theofanis et al. and Lawton et al. reported, an “unruptured” state seems more critical than AVM size alone in determining the extent of the post-operative deficit. One of the possible explanations for this is the need to conduct a more extensive dissection in the healthy portion of the brain [36,37]. The unruptured subgroups’ outcomes were better than those of the comparable ARUBA subgroups. Compared to ARUBA Spetzler–Martin Grades I and II lesions, the risk of persistent new deficit was lowest in unruptured Spetzler–Martin Grades I and II lesions. There was a higher incidence of persistent new deficits in outcomes for ARUBA Spetzler–Martin Grade III lesions and unruptured Spetzler–Martin Grade III lesions. A similar finding was reported by Johannes Schramm [15].

### 4.6. Factors Associated with a Poor Outcome after Resection of AVM

In our study, a high presurgical mRS score (mRS ≥ 3) was significantly related to poor functional outcomes, as shown in Table 3. Poor initial mRS (*p* < 0.0001) was linked to a poor clinical outcome in the study conducted by da Coast et al. [38]. Yang et al. reported that a lower pretreatment mRS score was favorably related to improved functional outcomes in a study on posterior fossa AVMs. According to these data, a poor initial mRS before treatment suggests a poor neurological outcome [39]. In our surgical series, however, since most patients with poor presurgical mRS came with life-threatening hematomas, emergency surgery for hematoma evacuation was required for these individuals. Despite poor preoperative mRS scores, 59% of these individuals can have favorable outcomes after surgery. Xiangzeng Tong reported similar results [35]. We continue to recommend early surgical intervention for these patients who had a poor initial mRS, and we propose immediate emergency surgery for those who had hematomas that posed a threat to their lives, plus AVM resection at the same time.

According to the findings of our study, the increasing patient age is a strong indicator of a poor neurological outcome after excision of an AVM. Hashimoto et al. also found that age > 60 y. was associated with poorer microsurgical outcomes, with 70% (16/23) of patients achieving good to excellent results, defined as full work capability or independence in completing activities of daily living, compared to 83% (76/92) at age < 60 [40]. Burkhardt et al. [12] also showed the influence of age: 83% of patients aged 60 to 65 had favorable outcomes, compared to 60% of those over 65 [41]. In addition, advancing age is associated with the decreased flexibility often seen in pediatric patients that contributes to improved neurological recovery after AVM microsurgery [42].

Because AVMs manifest congenital vascular malformation, advanced age increases the risk of eventual bleeding complications. The early resection of arteriovenous malformations (AVMs) following their first diagnosis is something that we generally recommend. In our surgical series, we also observed that being of female gender is another predictor of poor outcomes after arteriovenous malformation resection.

### 4.7. Study Limitations 

In this study, we examined the outcomes of microsurgery on 169 individuals who had arteriovenous malformations. There is a possibility of selection bias due to the study’s retrospective nature, a small sample size, the single-center experience, the patient referral pattern, and the preference of neurosurgeons. Since most studies on AVMs include microscopic, embolization, stereotactic radiosurgery (SRS), or a combination of the two, we only describe the functional outcomes of those with microsurgical resection of their AVMs without combining additional forms of treatment. Finally, this study’s findings lacked a longer follow-up time. A larger sample size and multi-center study are required to determine the outcomes of microsurgical treatment.

## 5. Conclusions 

The data from our study suggest that good outcomes can be obtained by microsurgical resection in most patients with AVMs. Independent predictors of poor outcomes after surgical resection of arteriovenous malformations (AVMs) include increasing age at the time of surgery, poor presurgical functional status, and being of female gender. Supposing that patients are more suitable for microsurgery after presurgical examination, outcomes usually are better in that case than those achieved by multimodal interventions (such as conservative treatment or ARUBA treatment arm). Therefore, we recommend early surgical removal of all surgically accessible AVMs to prevent successive hemorrhages and the consequence of poor neurological outcomes.

## Figures and Tables

**Figure 1 brainsci-13-01183-f001:**
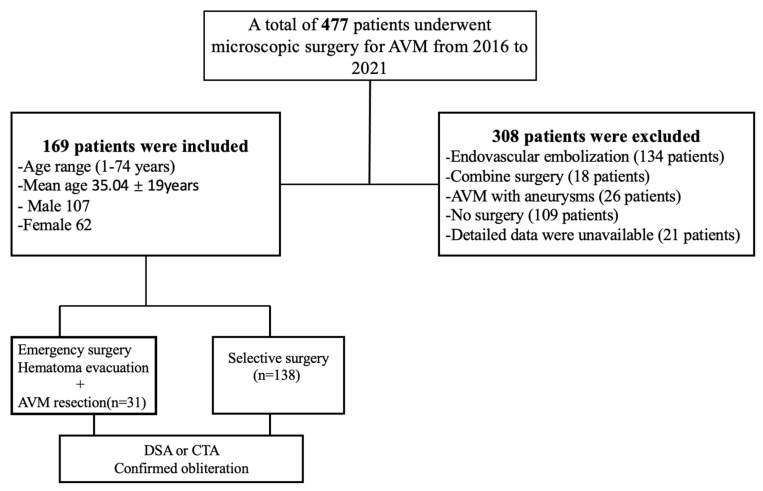
Flowchart illustrating the general enrollment process of patients with arteriovenous malformation (AVM). DSA, digital subtracting angiography; CTA, computed tomography angiography.

**Figure 2 brainsci-13-01183-f002:**
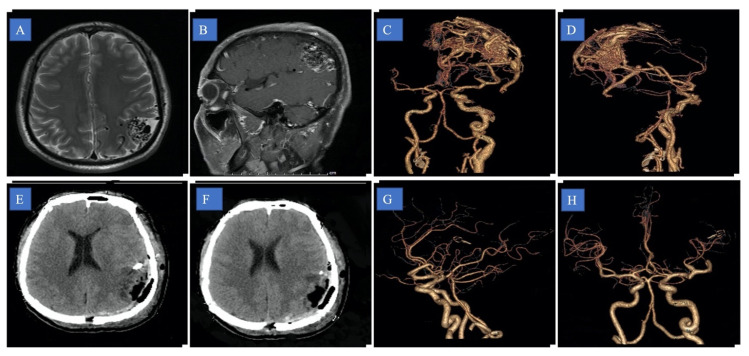
A 46-year-old man presenting with parietal cortex arteriovenous malformation (AVM), vertigo, and nausea. (**A**) Axial and (**B**) sagittal magnetic resonance image showing parietal cortex AVM; (**C**,**D**) pre-operative computed tomography angiography; (**E**,**F**) post-operative computed tomography; (**G**,**H**) post-operative computed tomography angiography.

**Figure 3 brainsci-13-01183-f003:**
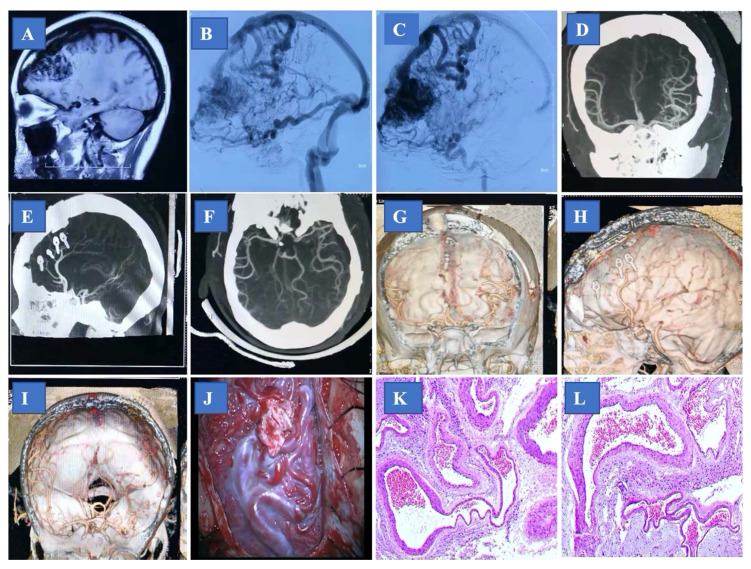
A 29-year-old man presenting with frontal lobe arteriovenous malformation (AVM), with severe headache. (**A**) Sagittal magnetic resonance image showing frontal lobe AVM; (**B**,**C**) pre-operative digital subtracting angiography; (**D**–**F**) post-operative computed tomography angiography; (**G**–**I**) post-operative computed Tomography Angiography three-dimensional reconstruction; (**J**) photo during the surgery; (**K**,**L**) histological examination with hematoxylin and eosin staining.

**Table 1 brainsci-13-01183-t001:** Frequency and percentage distribution of patients.

Variables	Number of Patients, (%)
Patients	169
Male	107 (63.7)
Female	62 (36.7)
Mean (range) age, years (±SD)	35.04 ± 19 (1–74)
BMI	
<18.5	33 (19.5)
18.5 ≤ 25	97 (57.4)
25 ≤ 30	34 (20.1)
>30	5 (3)
Blood group	
A	59 (34.9)
B	43 (25.4)
AB	16 (9.5)
O	51 (30.2)
Size, cm	
<3	107 (63.3)
≥3	62 (36.7)
Location	
Frontal lobe	41 (24.3)
Temporal lobe	45 (26.6)
Parietal lobe	12 (7.1)
Occipital lobe	27 (16)
Cerebellar	30 (17.8)
Frontoparietal lobe	5 (3)
Frontotemporal lobe	6 (3.6)
Parieto-occipital lobe	3 (1.8)
S–M grade	
I	65 (38.5)
II	46 (27.2)
III	32 (18.9)
IV	22 (13)
V	4 (2.4)
Presentation	
Headache	73 (43.2)
Neurological dysfunction	20 (11.8)
Seizures	14 (8.3)
Loss of consciousness	17 (10.1)
Vision problems	8 (4.7)
Dizziness and vomiting	30 (17.8)
Memory loss	2 (1.2)
Coma	5 (3)
Hemorrhage	
Yes	71 (42)
No	98 (58)
Duration of hospital stay	
<2 weeks	111 (65.7)
>2 weeks	58 (34.3)
Arterial supply	
Single	134 (79.3)
Double	35 (20.7)

BMI: body index mass; S–M grade: Spetzler–Martin grade.

**Table 2 brainsci-13-01183-t002:** Patients with good or poor outcomes after microsurgical AVM resection: a comparison of patient characteristics and AVM features.

Patient Parameter	All (*n* = 169)	Good Outcome (*n* = 145)	Poor Outcome (*n* = 24)	*p*-Value
Age	35.04 ± 19.01	33.63 ± 18.30	43.54 ± 21.33	0.017
Sex				
Male	112 (66.3)	102 (70.3)	10 (41.7)	0.006
Female	57 (33.7)	43 (29.7)	14 (58.3)	
BMI				
<18.5	33 (19.5)	27 (18.6)	6 (25.0)	0.384
18.5 ≤ 25	97 (57.4)	86 (59.3)	11 (45.8)	
25 ≤ 30	34 (20.1)	27 (18.6)	7 (29.2)	
>30	5 (3.0)	5 (3.4)	0(0)	
Blood group				
A	59 (34.9)	48 (33.1)	11 (45.8)	0.403
B	43 (25.4)	40 (27.6)	3 (12.5)	
AB	16 (9.5)	14 (9.7)	2 (8.3)	
O	51 (30.2)	43 (29.7)	8 (33.3)	
Location				
Frontal lobe	41 (24.3)	34 (23.4)	7 (29.2)	0.944
Temporal lobe	45 (26.6)	38 (26.2)	7 (29.2)	
Parietal lobe	12 (7.1)	10 (6.9)	2 (8.3)	
Occipital lobe	27 (16.0)	24 (16.6)	3 (12.5)	
Cerebellar	30 (17.8)	26 (17.9)	4 (16.7)	
Parietal-frontal lobe	5 (3.0)	4 (2.8)	1 (4.2)	
Frontal-temporal lobe	6 (3.6)	6 (4.1)	0 (0)	
Parietal-occipital lobe	3 (1.8)	3 (2.1)	0 (0)	
Size (cm)				
<3	107 (63.3)	93 (64.1)	14 (58.3)	0.585
≥3	62 (36.7)	52 (35.9)	10 (41.7)	
Presentation				
Headache	73 (43.2)	62 (42.8)	11 (45.8)	0.322
Neurological dysfunction	20 (11.8)	14 (9.7)	6 (25.0)	
Seizures	14 (8.3)	13 (9.0)	1 (4.2)	
Memory loss	2 (1.2)	2 (1.4)	0 (0)	
Loss of consciousness	17 (10.1)	15 (10.3)	2 (8.3)	
Vision problems	8 (4.7)	6 (4.1)	2 (8.3)	
Dizziness and vomiting	30 (17.8)	28 (19.3)	2 (8.3)	
Incidental	5 (3.0)	5 (3.4)	0 (0)	
Hemorrhage				
Yes	71 (42.0)	62 (42.8)	9 (37.5)	0.629
No	98 (58.0)	3 (57.2)	15 (62.5)	
Duration of hospital stay				
<2 weeks	111 (65.7)	91 (62.8)	20 (83.3)	0.049
>2 weeks	58 (34.3)	54 (37.2)	4 (16.7)	
mRS pre-surgery				
<3	99 (58.6)	78 (53.8)	21 (87.5)	0.002
≥3	70 (41.4)	67 (46.2)	3 (12.5)	
Duration of surgery (minutes)	283.71 ± 126.36	284.94 ± 126.61	276.25 ± 127.24	0.756
Blood loss	400.0 (225.0, 600.0)	400.0 (300.0, 600.0)	350.0 (200.0, 500.0)	0.237
Venous drainage				
Superficial drainage	108 (63.9)	87 (60.0)	21 (87.5)	0.009
Deep drainage	61 (36.1)	58 (40.0)	3 (12.5)	
Arterial supply				
Single	134 (79.3)	118 (81.4)	16 (66.7)	0.099
Double	35 (20.7)	27 (18.6)	8 (33.3)	
Recurrent rate				
No	164 (97.0)	141 (97.2)	23 (95.8)	0.706
Yes	5 (3.0)	4 (2.8)	1 (4.2)	
Follow-up time, month	41.01 ± 21.46	40.73 ± 21.99	42.71 ± 18.19	0.677

BMI: body index mass; mRS: modified Rankin Scale.

**Table 3 brainsci-13-01183-t003:** Risk factors for poor functional outcome after surgical resection of AVMs.

Variables	Univariate Analysis		Multivariate Analysis	
	OR (95% CI)	*p*-Value	OR (95%CI)	*p*-Value
Age	1.029 (1.006–1.052)	0.014	1.022 (1.000–0.045)	0.045
Sex				
Male	1 (Reference)		1 (Reference)	
Female	2.384 (1.055–5.386)	0.037	2.991 (1.309–6.832)	0.009
mRS pre-surgery				
<3	1 (Reference)		1 (Reference)	
≥3	0.159 (0.047–0.536)	0.003	0.206 (0.059–0.713)	0.013
Venous drainage				
Superficial	1 (Reference)		1 (Reference)	
Deep	0.237 (0.071–0.794)	0.02	0.318 (0.093–1.090)	0.068
Arterial supply				
Single	1 (Reference)		1 (Reference)	
Double	2.983 (1.271–6.998)	0.012	2.226 (0.934–5.306)	0.071

OR: Odds Ratio; CI: Confidence Interval; mRS: modified Rankin Scale.

**Table 4 brainsci-13-01183-t004:** All ARUBA-eligible patients’ data were compared to Spetzler–Martin Grades I and II, ARUBA-eligible patients.

Variable	All ARUBA-Eligible Patients	ARUBA-Eligible Spetzler–Martin Grades I and II Patients
Total	79	56
Sex		
Male	48 (60.8)	35 (62.5)
Female	31 (39.2)	21 (37.5)
Spetzler–Martin grade		
I	24 (30.4)	24 (42.9)
II	32 (40.5)	32 (57.1)
III	18 (22.8)	0 (0)
IV	5 (6.3)	0 (0)
V	0 (0)	0 (0)
Size		
<3	56 (70.9)	41 (73.2)
≥3	23 (29.1)	15 (26.8)
Symptoms		
Headache	40 (50.6)	29 (51.8)
Dizziness and vomiting	19 (24.1)	14 (25.0)
Seizure	11 (13.9)	8 (14.3)
Loss of consciousness	5 (6.3)	3 (5.4)
Vision problems	4 (5.1)	2 (3.6)
Location		
Temporal lobe	29 (36.7)	22 (39.3)
Frontal lobe	23 (29.1)	17 (30.4)
Cerebellar	17 (21.5)	10 (17.9)
Occipital	10 (12.7)	7 (12.5)
Venous drainage		
superficial	61 (77.2)	48 (85.7)
deep	18 (22.8)	8 (14.3)
Arterial supply		
single	68 (86.1)	49 (87.5)
double	11 (13.9)	7 (12.5)
Deficit		
New deficit	9 (11.4)	5 (8.9)
Persistent new deficit	3 (3.8)	2 (3.6)
Mortality	0	0

ARUBA = A Randomized Trial of Unruptured Brain Arteriovenous Malformations.

**Table 5 brainsci-13-01183-t005:** Deficits in the study population and its various subgroups, both new deficits and persistent new deficits.

Variables	All AVMs	All Unruptured AVMs	All Hemorrhagic AVMs	All ARUBA-Eligible AVMs	ARUBA-Eligible Spetzler–Martin Grades I and II
No. of patients	169	98	71	79	56
% New deficits	16.6	9.2	4.3	11.4	8.9
% Persistent new deficits	8.9	3.1	2.8	3.8	3.6
% Mortality	5.3	1	3.8	0	0

AVM, Arteriovenous malformation; ARUBA = A Randomized Trial of Unruptured Brain Arteriovenous Malformations.

**Table 6 brainsci-13-01183-t006:** All patients available for follow-up who had deficits regarding Spetzler–Martin grades.to Spetzler–Martin grade.

Spetzler–Martin Grade	No. of Patients	No. of New Deficits (%)	No. of Persistent New Deficits (%)
I	65	4 (6.2)	1 (1.5)
II	46	5 (10.9)	2 (4.3)
III	32	5 (15.6)	3 (9.4)
IV	22	10 (45.5)	7 (31.8)
V	4	3 (75)	2 (25)
Total	169	29	15

## Data Availability

The data that support the findings of this study are included in the article. Further inquiries are available from the corresponding author upon reasonable request.

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
