# Peer review of "Microsurgical Treatment of Arteriovenous Malformations: A Single-Center Study Experience"

_brainsci, 2023, doi:10.3390/brainsci13081183_

Round 1

Reviewer 1 Report

The authors present their cases series on microsurgically trated AVMs

The work althoug is not a novelty is good and the data are well presented

Author Response

The detailed response is included in the attached file. Please refer to that attached file. Thank you

Reviewer 2 Report

This paper sounds interesting. Some points need revision:

- Lines 30-31: "Our study suggests that better outcomes can be obtained by microsurgical resection in majority of patients with AVMs" Better than what? Revise this part.

- in "material and methods" it should be written how patients underwent surgery were selected? based on which parameters?

- Is there any difference in the outcome between elective patients undergoing surgery and those operated on urgently for bleeding? highlight this concept

- Line 310-314. In the discussion section, authors should add an historical part on AVM. Look at: -- Julius Caesar's Epilepsy: Was It Caused by A Brain Arteriovenous Malformation? World Neurosurg. 2015 ---- "Dual pathology" and the significance of surgical outcome in "Dostoewsky's epilepsy". Epileptic Disord. 2000

- Lines 345-350: I like this part. Consider this more recent paper: "Spetzler-Martin Grade III Arteriovenous Malformations: A Comparison of Modified and Supplemented Spetzler-Martin Grading Systems. Neurosurgery. 2021"

- Lines 234-236. Table 2. Age appears to greatly affect outcome. Improve and discuss more this part.

- Lines 464-468. What does this paper add new to the current literature? Is surgery still useful in patients with AVM? Highlight this concept.

Minor editing of English language required

Author Response

(The authors gave the same response as above.)

Reviewer 3 Report

Thank you for the opportunity to review this manuscript. This is an interesting study about the safety and efficacy of microsurgical resection of AV malformations. Although, there are a couple of issues, including the high risk of selection bias due to the design of the study, and the chance of type 1 error due to multiple comparisons, the authors have attempted to address these by conducting a multivariate analysis and acknowledging the limitations of their study design. These limitations may affect the generalizability of this study but the analysis still adds to the existing literature. I will recommend that authors edit the manuscript to improve the flow and decrease the redundancies in the manuscript. In the current iteration it a difficult read and will benefit from improved flow.

There are grammatical errors in the manuscript that need to be addressed. Also, the flow is unsteady and would benefit from revision.

Author Response

(The authors gave the same response as above.)

Reviewer 4 Report

In their article, the authors describe outcome after microsurgical treatment of AVMs with associated influencing factors.
The article is well written, clearly structured and easy to understand.
However, some minor points should be revised:

Abstract - optimistic outcome - The authors probably mean otpimal outcome.
Page 3 From line 95 - Surgery instead of Surgary
Please rephrase selective surgery to elective surgery
Typos like page 4, line 125 - analyses etc.
Please pay attention to excess spaces (e.g. heading baseline table, often throughout the text) and partly in the baseline table (number of patients)
Results: The authors mix up absolute numbers and percentages here (page 5, from line 136)
Page 7, line 197: What is meant by the statement (angiography in the following years - shortly after surgery?)
Page 7, from line 211: reasons for death?
Sequence of tables 4-6 not chronological
Page 10, section from line 250 - missing comma, semicolon at OR/CI.
Discussion: sources for the first section are missing

The article should be checked for typos and minor formatting errors (spaces).

Author Response

(The authors gave the same response as above.)

Round 2

Reviewer 2 Report

Authors solved all my criticisms.

 Minor editing of English language required